# Deconfounded Emotion Guidance Sticker Selection with Causal Inference

Jiali Chen
South China University of Technology
Guangzhou, China
segarychen@mail.scut.edu.cn

Yi Cai
South China University of Technology
Guangzhou, China
ycai@scut.edu.cn

Ruohang Xu
South China University of Technology
Guangzhou, China
202030482249@mail.scut.edu.cn

Jiexin Wang
South China University of Technology
Guangzhou, China
jiexinwang@scut.edu.cn

Jiayuan Xie*
The Hong Kong Polytechnic
University
Hong Kong, China
jiayuan.xie@polyu.edu.hk

Qing Li
The Hong Kong Polytechnic
University
Hong Kong, China
csqli@comp.polyu.edu.hk

## Abstract

With the increasing popularity of online social applications, stickers have become common in online chats. Teaching a model to select the appropriate sticker from a set of candidate stickers based on dialogue context is important for optimizing the user experience. Existing methods have proposed leveraging emotional information to facilitate the selection of appropriate stickers. However, considering the frequent co-occurrence among sticker images, words with emotional preference in the dialogue and emotion labels, these methods tend to over-rely on such dataset bias, inducing spurious correlations during training. As a result, these methods may select inappropriate stickers that do not match users' intended expression. In this paper, we introduce a causal graph to explicitly identify the spurious correlations in the sticker selection task. Building upon the analysis, we propose a Causal Knowledge-Enhanced Sticker Selection model to mitigate spurious correlations. Specifically, we design a knowledge-enhanced emotional utterance extractor to identify emotional information within dialogues. Then an interventional visual feature extractor is employed to obtain unbiased visual features, aligning them with the emotional utterances representation. Finally, a standard transformer encoder fuses the multimodal information for emotion recognition and sticker selection. Extensive experiments on the MOD dataset show that our CKS model significantly outperforms the baseline models.

## CCS Concepts

• **Information systems** → **Multimedia content creation**.

## Keywords

Sticker Selection, Emotion Recognition, Causal Inference

**ACM Reference Format:**
Jiali Chen, Yi Cai, Ruohang Xu, Jiexin Wang, Jiayuan Xie, and Qing Li. 2024. Deconfounded Emotion Guidance Sticker Selection with Causal Inference.

*Corresponding author.

MM '24, October 28-November 1, 2024, Melbourne, VIC, Australia
© 2024 Copyright held by the owner/author(s).
ACM ISBN 979-8-4007-0686-8/24/10
https://doi.org/10.1145/3664647.3681522

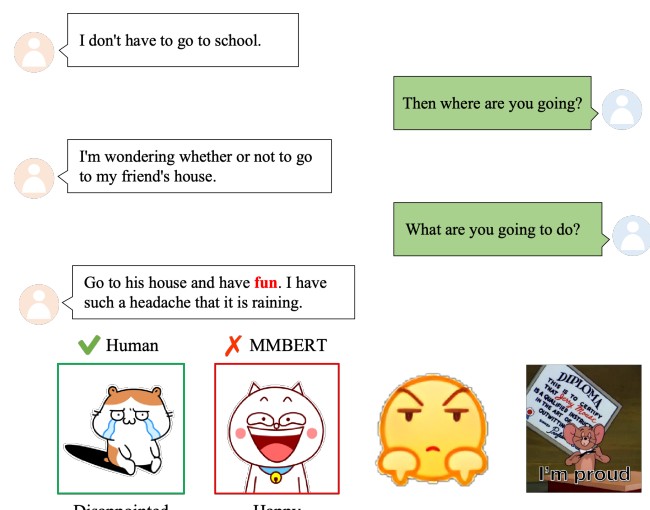

**Figure 1: The sample from MOD dataset about the spurious correlations in the sticker selection task. Words below sticker images indicate the predicted emotion.**

In *Proceedings of the 32nd ACM International Conference on Multimedia (MM '24), October 28-November 1, 2024, Melbourne, VIC, Australia.* ACM, New York, NY, USA, 10 pages. https://doi.org/10.1145/3664647.3681522

## 1 Introduction

With the widespread use of online social applications (e.g. WeChat, Instagram and X), stickers have become a prevalent way for individuals to convey their attitudes and emotions in online communication. These graphical elements of stickers play an important role in enhancing users' perceptions of intimacy, positivity, and social connectedness [21, 40]. Thus, an effective sticker recommendation system holds the promise of substantially assisting users in expressing their emotions vividly and conveying their intended messages, leading to more engaging and meaningful interactions in online communication [2, 14, 23].

Toward this end, a number of models [15, 27, 44] have emerged to tackle the task of sticker selection, aiming for an effective recommendation. It is the task of selecting the appropriate sticker

from a candidate sticker set based on the dialogue context. Upon scrutinizing these models, they typically focus on aligning the text and sticker image information through sophisticated mechanisms, overlooking the essential emotional information in the process of sticker selection. Previous studies [22, 49] point out that excessive reliance on vision-language alignment is not inherently the most critical aspect. Instead, it's more important to uncover and utilize emotional information when selecting stickers. For example, a sticker depicting a cat or a dog smiling generally conveys the emotion of happiness, independent of the particular animal species in stickers. Specifically, Zhang et al. [49] attempt to incorporate emotion into the sticker selection task with a multitask learning framework. However, this method straightforwardly implements the emotion classification and always learns the frequent co-occurrence among the sticker image, words with emotional preference and emotion labels in the dataset. For instance, it is common to encounter scenarios where the presence of the word "fun" leads to the selection of a smiling-themed sticker associated with happy emotion. It might harmfully mislead the model to automatically lean towards selecting stickers with smiles whenever the word "fun" appears, as illustrated in Fig. 1. This previously inherent dataset bias causes the model to over-exploit the spurious correlations among the sticker image, words with emotional preference and emotion. Consequently, it is essential to analyze and alleviate the bias in the sticker selection task.

In light of pedagogy [41], the above limitations in the sticker selection model can be partially attributed to the model's over-dependence on intuitive processing, rendering the models prone to various decision biases. The behavior of inappropriate sticker selection is referred to as the "heuristic system" [41]. To address the limitation, human typically relies on an "analytic system" [41] to select the appropriate sticker associated with emotion. Specifically, inspired by the selectivity [22] in human communication behavior, the "analytic system" first pre-processes to find out the emotional utterance within the dialogue context since some utterances offer scarcely emotional information relevant to the sticker selection. As shown in Fig. 1, the utterance "I am wondering whether or not to go to my friend's house?" reflects hesitated emotion. At this step, we should also extract utterances that convey emotional information, even explicit emotional words are absent within the utterances. Subsequently, the system engages in systematic information processing, allowing it to accurately evaluate and interpret the emotional cues within the dialogue and sticker image. It can mitigate the impact of various decision biases but requires more mental resources. For example, our focus should not be confined to words with an emotional preference, such as "fun", as illustrated in Fig. 1. Instead, we should prioritize the accurate emotional information in the dialogue.

In this paper, we present a causal graph [35] to analyze the cause of the spurious correlations in sticker selection. Guided by this analysis, we propose the Causal Knowledge-Enhanced Sticker Selection (CKS) model to emulate "pre-process" and "process" steps in the analytic system, which consists of three components: a knowledge-enhanced emotional utterance extractor (KEUE), an interventional visual feature extractor (IVE), and a multimodal encoder. For the "pre-process" step, the KEUE is devised to identify emotional utterances within the dialogue. Considering that the emotional and non-emotional utterances are not annotated in the dialogue, we

involve the incorporation of six types of commonsense knowledge generated by ATOMIC [39] model and utilize utterances with generative commonsense knowledge of causal relationship to assist in identifying emotional utterances. Moreover, certain words with emotional preference often appear in emotional utterances, potentially leading the model to select the inaccurate sticker, as illustrated in Fig. 1. Thus, for the "process" step, the IVE incorporates causal inference to extract unbiased visual features of the input sticker image and align them with the emotional utterances representation. Specifically, we first design a visual knowledge deconfounder (VKD), which aims to mitigate the negative effect of words with emotional preference. Given the stylistic diversity of stickers, we introduce a content invariant module (CIM) to separate the content variable from the style variable, thereby enhancing the generalization capability for the sticker selection. The objective of IVE is to align unbiased visual features with the emotional utterances representation. Finally, a standard transformer encoder [42] is adopted to fuse the multimodal input information (i.e., dialogue context, emotional utterances representation, textual information in the sticker and unbiased visual features) to select the sticker.

Our contributions can be summarized as follows:

- To the best of our knowledge, we are the first to investigate the inherent bias in the sticker selection task from a causal inference perspective and analyze that such bias induces the model to learn such spurious correlations, leading to incorrect sticker selection.
- Our proposed Causal Knowledge-Enhanced Sticker Selection (CKS) model adopts a causality perspective to effectively mitigate the issue of the above bias. To achieve this goal, it leverages commonsense knowledge to identify emotional utterances and extract unbiased visual features for alignment, streamlining the sticker selection process.
- Extensive experiments on MOD [13] dataset validate the superiority of our CKS model in mitigating the spurious correlations and recognizing accurate emotion during sticker selection. In particular, CKS achieves substantial performance improvement and significantly surpasses baseline models.

## 2 Related Work

### 2.1 Sticker Selection

Previous studies have primarily focused on emoji recommendation in multimodal dialogue systems [3, 4, 46]. However, in comparison to stickers, emojis are inherently limited in variety and expression. In recent years, some works have explored the sticker selection task [13, 15, 27, 44, 49]. For instance, Laddha et al. [27] employ a clustering method to predict messages and then substitute them with stickers. Gao et al. [15] adopt a co-attention matrix to calculate the attention between utterance and sticker representation. They further apply a deep interaction network to fuse the multimodal information for image-text matching. Fei et al. [13] utilize the pre-trained GPT-2 [38] model to jointly process the information in the dialogue (i.e., stickers and utterances) for sticker selection. Zhang et al. [49] propose a multimodal BERT [8] with a multitask learning method to combine multimodal input information. Specifically, three auxiliary tasks (i.e., masked context prediction, sticker semantic prediction and sticker emotion classification) are proposed to enhance the understanding of dialogues and stickers.

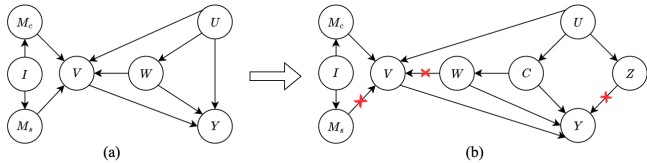

**Figure 2: Causal graph of sticker selection. The solid arrow
(→) denotes causation between variables.**

## 2.2 Causal Inference

Causal inference is an effective analytical tool to measure the causal
effect between two variables, rather than depending only on their
statistical correlation [34, 35]. Several approaches attempt to in-
corporate causal inference with deep neural networks (DNNs) in
the field of natural language processing (NLP) [12, 30, 45] and com-
puter vision (CV) [7, 36, 43]. Specifically, Wang et al. [43] propose a
Visual Commonsense R-CNN network, which integrates causal in-
tervention [35] to predict contextual objects for image caption and
visual question answering. Wei et al. [45] leverage counterfactual
inference to model the fine-grained user preference in the knowl-
edge graph based recommendation. Yuan et al. [47] identify the
concept bias in the concept extraction system. Then, they equip the
knowledge-guided prompts as an intervention for the pre-trained
language model to mitigate the bias. Chen et al. [7] mitigate the
bias in object attributes and relationships for visual question gener-
ation. To the best of our knowledge, we are the first to analyze the
spurious correlations in the sticker selection task and address this
problem from a causal view.

## 3 Causal View at Sticker Selection

### 3.1 Causal Graph of Sticker Selection

We first formalize the causality to accurately identify the multi-
modal emotion for the sticker selection task with a causal graph [35],
as shown in Fig. 2 (a). The cause-and-effect relationships among
seven variables: dialogue context $U$, candidate sticker images $I$,
sticker visual knowledge $V$, style variable $M_s$ of the sticker images,
content variable $M_c$ of the sticker images, words with emotional
preference $W$ in the dialogue and current emotion $Y$ of the speaker
when selecting the sticker. We explain the relationships in detail:
**i)** $U \rightarrow Y$ and $M_c \leftarrow I \rightarrow M_s$. Intuitively, the emotion conveyed
by the speaker when selecting a sticker is influenced by the the
dialogue context. Since stylistic differences among stickers, the
sticker image contains the content and style variables. **ii)** $U \rightarrow V$
and $M_c \rightarrow V \leftarrow M_s$. The visual knowledge is obtained through a
series of attention operations on dialogue context, content and style
variables (i.e., $M_c$ and $M_s$) from the sticker image. **iii)** $U \rightarrow W \rightarrow V$.
The confounder includes words with specific emotional preference
from the dialogue (i.e., $U \rightarrow W$), as depicted in Fig. 1, where "fun"
serves as the word with emotional preference. Additionally, the
visual knowledge of the sticker is frequently mentioned in the
word with emotion preference, which establishes a direct shortcut
between them (i.e., $W \rightarrow V$). For example, the model selects the
sticker with visual knowledge of a smile based on the word with
emotional preference "fun" in Fig. 1. **iv)** $W \rightarrow Y$ and $V \rightarrow Y$. The
causal effect exists since the words with emotional preference also

impact the probability outputs of the emotion classifier. The emo-
tion $Y$ is conditional upon the visual knowledge $V$, showcasing the
impact of extracted visual information on the emotional expres-
sion within the dialogue context. The process of sticker selection is
intrinsically related to the speaker's emotion $Y$. Therefore, as the
causal graph in Fig. 2 (a), we should focus on the accurate emotion.

### 3.2 Spurious Correlations

Taking a closer look at the causal graph in Fig. 2 (a), we first ana-
lyze the spurious correlations present in the sticker selection task.
Given the fact that only a portion of the dialogue context is useful
for emotion when selecting the sticker of interest, while the rest
offers scarcely information relevant to the sticker selection [22].
Moreover, the words $W$ with specific emotion preference frequently
exist in the emotional utterance, which also serve as the confounder
between visual knowledge $V$ and $Y$. As a result, the backdoor path
$V \leftarrow W \rightarrow Y$ becomes available, causing spurious correlations.
Worse still, previous models have difficulty learning robust repre-
sentations of stickers in various style distributions [16, 20].

Building on the above analysis, previous models frequently learn
such spurious correlations, leading to the misidentification of emo-
tions. We design a causal graph as shown in Fig. 2 (b) to mitigate
spurious correlations. Specifically, these models often struggle to
distinguish between emotional utterances and non-emotional ut-
terances due to their ineffective modeling of the dialogue context.
Inspired by the selectivity in human communication behavior [22],
we design a knowledge-enhanced emotional utterance extractor
(KEUE) to effectively mitigate such spurious correlations, which
can effectively leverage the emotional utterances from the dialogue
context as variable $C$ and avoid the effect from the non-emotional
utterances $Z$. Furthermore, since the existence of the backdoor
path $V \leftarrow W \rightarrow Y$, the model may be biased toward the visual
knowledge related to the words with emotional preference, while
overshadowing its perception of the broader visual context within
the images. The model might emphasize the localized visual cues
tied to specific words for emotion recognition. To eliminate this
confounding problem, we use the backdoor adjustment [35] by
controlling all possible values of $W$. For the distribution shift in
the style variable across different sticker images, we decouple con-
tent representation from style representation with causal inference,
which can enhance the model's generalization capability under
various sticker styles.

## 4 Methodology

Formally, given $N$ candidate stickers $S = \{s_1, s_2, ..., s_N\}$, the sticker
selection task aims to select the appropriate sticker $\hat{s}$ based on the
multi-turn dialog context $U = \{u_1, u_2, ..., u_T\}$ between two speak-
ers, where $T$ represents the number of utterances in the dialogue
and $u_T$ is the last utterance. It requires a precise comprehension of
the dialogue context, accurate emotion prediction and a convincing
rationale for the sticker selection. Getting inspiration from the "an-
alytic system" [41] in pedagogy, we consider the sticker selection
task from a causal perspective and propose our Causal Knowledge-
Enhanced Sticker Selection (CKS) to imitate the "pre-process" and
"process" steps in the "analytic system" subsequently for debiasing.
The architecture of the CKS model is illustrated in Fig. 3, which

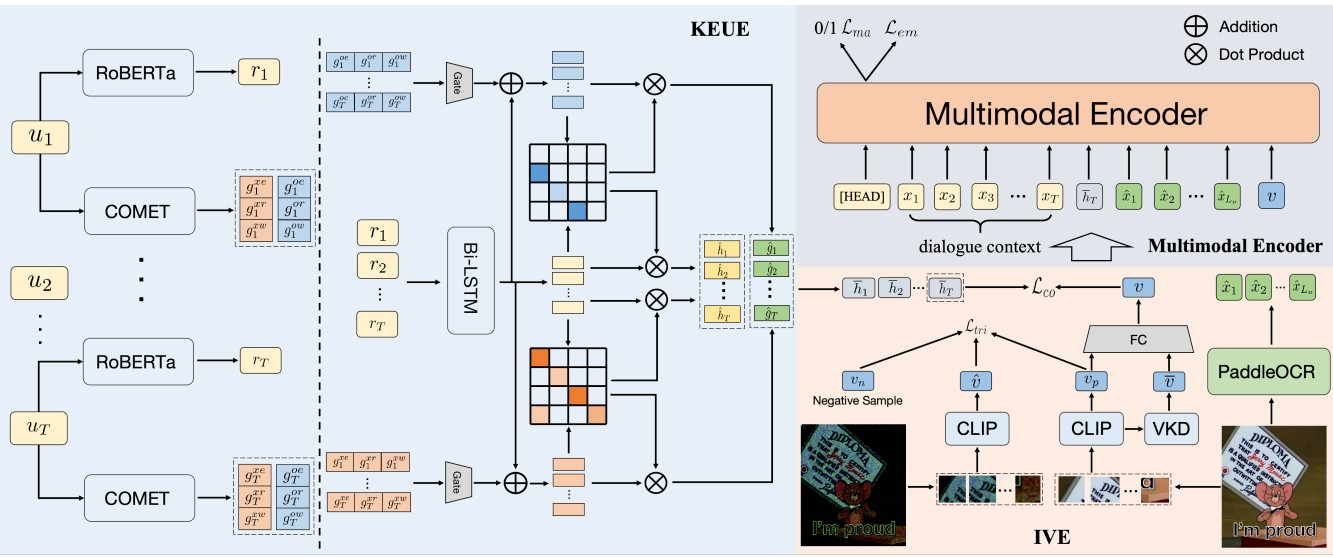

**Figure 3: Overview of our CKS model. It contains three components: (i) the knowledge-enhanced emotional utterance extractor (KEUE), (ii) the interventional visual feature extractor (IVE), where VKD represents visual knowledge deconfounder in section 4.2.1, and (iii) the multimodal encoder.**

consists of three components: (i) knowledge-enhanced emotional utterance extractor (KEUE), which imitates the "pre-process" step and aims to identify emotional utterances with commonsense knowledge to facilitate further sticker selection. (ii) interventional visual feature extractor (IVE), which incorporates causal inference to disentangle the visual features of stickers, and obtain unbiased visual features to align with the emotional utterances representation from the knowledge-enhanced emotional utterance extractor for the "process" step. (iii) multimodal encoder, which utilizes a standard transformer encoder [42] to process multimodal information (i.e., dialogue context, emotional utterances representation, OCR features and unbiased visual knowledge). The objective of this component is to distinguish the positive and negative stickers and recognize the emotion when selecting the sticker. The details of each component are in the following subsections.

### 4.1 Knowledge-Enhanced Emotional Utterance Extractor

Considering the selectivity [22] in human communication behavior, humans often concentrate on specific emotional utterances within the dialogue, selectively ignoring other pieces of dialogue information. The knowledge-enhanced emotional utterance extractor (KEUE) aims to distinguish emotional utterances from non-emotional utterances within the dialogue context.

*4.1.1 Dialogue Encoding.* We first insert '[CLS]' and '[speaker $j$]' ($j$=1,2) tokens in the beginning of each utterance $u_i$ and utilize a pre-trained RoBERTa [31] model to obtain the contextual features of '[CLS]' token $\{r_i\}_{i=1}^T$, where $r_i \in \mathbb{R}^{d^r}$ and $d^r$ is the dimension of the utterance-level representation. We aim to model sequential dependencies between successive utterances with the temporal information embedded in the entire dialogue context. Specifically, we adopt a Bi-directional Long Short-Term Memory (bi-LSTM) [19]

to extract the conversational representation of each utterance:

$$\overrightarrow{h_i} = \overrightarrow{\text{LSTM}}\left(r_i, \overrightarrow{h_{i-1}}\right), \quad \overleftarrow{h_i} = \overleftarrow{\text{LSTM}}\left(r_i, \overleftarrow{h_{i-1}}\right), \quad (1)$$

where $\overrightarrow{h_i} \in \mathbb{R}^d$ and $\overleftarrow{h_i} \in \mathbb{R}^d$ are the hidden state of the $i$-th utterance for the forward and backward of LSTM, respectively. Next, we obtain the representation of the $i$-th utterance $h_i = [\overrightarrow{h_i}; \overleftarrow{h_i}]$, where $h_i \in \mathbb{R}^{2d}$ and $[;]$ denotes the concatenation operation.

*4.1.2 Knowledge Acquisition.* Due to the lack of available annotated emotional utterances, we cannot directly calculate the effect of $U \rightarrow C \rightarrow Y$ in Fig. 2 (b). Previous studies [6, 39] demonstrate neural networks can utilize the commonsense knowledge to predict probable causes-effects of previously unseen events and underlying emotion. We treat previous utterances along with commonsense knowledge as emotional utterances $C$.

In our work, we utilize ATOMIC [39] as the commonsense knowledge base due to its vast everyday inferential knowledge. Specifically, we investigate six types of commonsense knowledge from ATOMIC, as shown in Fig. 4. To provide further clarity, "xReact", "xEffect" and "xWant" knowledge types are for intra-emotional utterances, which signify the influences or outcomes resulting from utterances within the same speaker in the dialogue. Additionally, "oReact", "oEffect", and "oWant" knowledge types are for inter-emotional utterances, indicating the effects exerted on others or what others would like to do and feel after receiving the current utterance. To acquire the commonsense knowledge from the ATOMIC, we utilize the generative commonsense transformer model (i.e., COMET [5]) to generate commonsense knowledge, which is pre-trained on ATOMIC. Specifically, we first concatenate the $i$-th utterance in the dialogue with the $k$ type of commonsense knowledge $R_k$ as input sequence $(u_i, R_k, [\text{GEN}])$ for COMET model,

where [GEN] is a special token for commonsense knowledge generation, $k \in \{xe, xr, xw, oe, or, ow\}$ and $xe, xr, xw, oe, or, ow$ are short for knowledge types "xEffect", "xReact", "xWant", "oEffect", "oReact" and "oWant", respectively. Next, we use the BART-based [28] variation of COMET to generate inferential commonsense knowledge for the $i$-th utterance under the knowledge type $R_k$ and obtain the features from the last hidden state as knowledge representation $g_i^k = \text{COMET}(u_i, R_k, [\text{GEN}])$, where $g_i^k \in \mathbb{R}^{1024}$. We then design gating mechanisms to learn the enriched knowledge representation for intra- and inter-speaker:

$$
\begin{aligned}
g_i^a &= \text{sigmoid}([g_i^{xe} + g_i^{xr} + g_i^{xw}]W_a + b_a), \\
g_i^e &= \text{sigmoid}([g_i^{oe} + g_i^{or} + g_i^{ow}]W_e + b_e),
\end{aligned}
\tag{2}
$$

where $g_i^a$ and $g_i^e$ represent the knowledge representation for intra- and inter-speaker in the $i$-th utterance, respectively. The sigmoid activation function is denoted by $\text{sigmoid}(\cdot)$. $W_m$ and $W_e \in \mathbb{R}^{1024 \times 2d}$, $b_a$ and $b_e$ are learnable parameters.

*4.1.3* **Emotional Utterances Representation**. Our goal is to extract the intra- and inter-emotional utterances based on the dialogue context with commonsense knowledge. Specifically, the attention scores $s_{i,j}^{\beta}$ ($\beta = a, e$) is used to measure the relevance between the $i$-th and $j$-th utterances with commonsense knowledge:

$$
\begin{aligned}
s_{i,j}^a &= \text{softmax}(F_q(h_i)[(F_k(h_j) + F_v(g_j^a))]^T \cdot m_{i,j}^a), \\
s_{i,j}^e &= \text{softmax}(F_q(h_i)[(F_k(h_j) + F_v(g_j^e))]^T \cdot m_{i,j}^e),
\end{aligned}
\tag{3}
$$

where $s^a$ and $s^e$ is the attention scores from the intra- and inter-speaker repestively. The $F_q, F_k, F_v$ are linear layers. Meanwhile, the mask values $m^a$ and $m^e$ ensure the temporal sequence and correctness during dialogue modeling.

Once we obtain the attention scores for the emotional utterances, the representation of the $i$-th utterance can be formulated as:

$$
\hat{h}_i = \sum_{j \in S(i)} s_{i,j}^a F_q(h_j) + \sum_{j \notin S(i)} s_{i,j}^e F_q(h_j),
\tag{4}
$$

where $\hat{h}_i \in \mathbb{R}^{d^r}$ and $S(i)$ represents the set of utterances delivered by the same speaker as the $i$-th utterance in the dialogue. Moreover, it is also essential to consider the $i$-th utterance with commonsense knowledge information:

$$
\hat{g}_i = \sum_{j \in S(i)} s_{i,j}^a (F_k(h_j) + F_v(g_j^a)) + \sum_{j \notin S(i)} s_{i,j}^e (F_k(h_j) + F_v(g_j^e)), \tag{5}
$$

where $\hat{g}_i \in \mathbb{R}^{d^r}$. We concatenate $\hat{h}_i$ with $\hat{g}_i$, then proceed through a Multilayer Perceptron (MLP) $\text{MLP}_{hg}$ to obtain the emotional utterances representation for the $i$-th utterance:

$$
\bar{h}_i = \text{MLP}_{hg}([\hat{h}_i; \hat{g}_i]),
\tag{6}
$$

where $\bar{h}_i \in \mathbb{R}^{d^r}$ and $[;]$ denotes the concatenation operation.

## 4.2 Interventional Visual Feature Extractor

The interventional visual feature extractor aims to effectively mitigate the spurious correlations shown in Fig. 2 from a visual perspective. Its objective is to procure unbiased visual features that align with the representation of emotional utterances. Technically, we first use the same setup as [49] to create a negative sticker sample for each dialogue during training and ensure the emotional content

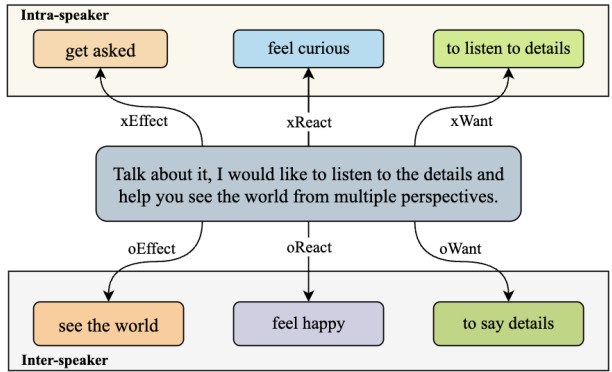

**Figure 4: An utterance with commonsense knowledge.**

of the negative image is inconsistent with that of the positive one. Then, we use the CLIP [37] visual encoder for sticker image feature extraction. Specifically, we use the embedding of the '[CLS]' token as the representation of the whole image features for positive and negative images, denoted as $v_p$ and $v_n$, respectively, where $v_p, v_n \in \mathbb{R}^{d^r}$. Finally, we design two modules: visual knowledge Deconfounder (VKD) and content invariant module (CIM) as depicted in Fig. 5, to prevent learning spurious features and preserve transferable invariant content features of the sticker image.

*4.2.1* **Visual Knowledge Deconfounder**. Upon inspection on the causal graph in Fig. 2, the confounder $W$ between $V$ and $Y$ opens the backdoor path $V \leftarrow W \rightarrow Y$ making them spuriously correlated. Therefore, we devise a visual knowledge deconfounder (VKD) to eliminate these spurious correlations using backdoor adjustment [35]. The overview of the VKD module is shown in Fig. 5. Specifically, we implement a causal intervention on variable $V$ (i.e., $P(Y \mid do(V))$) to block the backdoor path, where $do(\cdot)$ operator is employed to cut $W \rightarrow V$ with backdoor adjustment:

$$
P(Y \mid do(V)) = \sum_i P(w_i)P(Y \mid V, w_i),
\tag{7}
$$

where $w_i$ is the $i$-th sample in confounder dictionary $W$. Considering that the words with emotional preference cannot be directly accessed, we adopt the widely recognized six emotion categories proposed by Ekman [11] (i.e., surprise, happiness, disgust, fear, sadness, anger) to form the confounder dictionary. Specifically, we utilize a BERT [8] model to process these emotion categories.

By utilizing the Normalized Weighted Geometric Mean (NWGM) approximation, Eq. 7 can be approximated as:

$$
\bar{v} = \text{softmax}(\text{FC}((v_p^T W)W)),
\tag{8}
$$

where $\bar{v} \in \mathbb{R}^{d^r}$ is visual features from VKD and $\text{FC}(\cdot)$ is a fully-connected layer.

*4.2.2* **Content Invariant Module**. Considering that the sticker image contains the content variable $M_c$ and style variable $M_s$, we argue that the content of the sticker when selecting it is independent of the style variable [32, 33]. Therefore, we aim to separate the content representation from the style representation with causal inference. Based on the analysis, the content variable $M_c$ is expected to remain invariant over the randomization on the style variable $M_s$

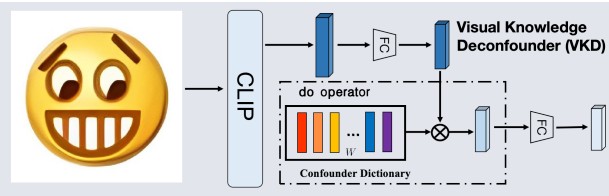

**Figure 5: Visual knowledge deconfounder (VKD) module.**

(i.e., $P(M_c \mid do(M_s))$). Building upon the insight that the extremely high and low frequency components of images often contain more style-specific features [18], we can implement the randomization of these components to simulate intervention on the style variable. Specifically, we first utilize a discrete cosine transform [1] $\mathcal{F}(\cdot)$ to obtain the frequency spectrum of the input sticker image $I$ (i.e., $\mathcal{F}(I)$). We adopt the band-pass filter $\mathcal{H}(\cdot)$ to distinguish between the content variable and style variable:

$$\mathcal{H}(R_h, R_l) = e^{-\frac{a^2+b^2}{2R_h^2}} - e^{-\frac{a^2+b^2}{2R_l^2}}, \tag{9}$$

where $(a, b)$ is the positions of the spectrum. $R_h$ and $R_l$ denote the low and high frequency threshold, respectively. Next, a Gaussian distribution $\text{Gus}(I) = I \cdot (1 + \mathcal{N}(0, 1))$ is employed to randomize the style variable, $\mathcal{N}(0, 1)$ denotes the standard Gaussian distribution with a mean of 0 and a standard deviation of 1. We generate an augmented image $\hat{I}$ with inverse discrete cosine transform $\widehat{\mathcal{F}}(\cdot)$:

$$\hat{I} = \widehat{\mathcal{F}}(\text{Gus}(\mathcal{H}(R_h, R_l) \cdot \mathcal{F}(I)) + (1 - \mathcal{H}(R_h, R_l)) \cdot \mathcal{F}(I)). \tag{10}$$

Subsequently, we also employ the CLIP visual encoder to extract features from the augmented sticker image as $\hat{v} \in \mathbb{R}^{d^r}$. The contextual features of '[CLS]' token represent the image features. To simulate the process of content invariant with intervention over style (i.e., $P(M_c \mid do(M_s))$), we optimize the CLIP visual encoder with the standard triplet loss to focus on the content variable during the above intervention. The triplet loss can be formulated as:

$$\mathcal{L}_{tri} = max\{0, \mu + \text{Dis}(v_p, \hat{v}) - \text{Dis}(v_p, v_n)\}, \tag{11}$$

where $\mu$ denotes a margin and the distance function is defined as $\text{Dis}(v_1, v_2) = \frac{1 - v_1 \cdot v_2}{||v_1|| - ||v_2||}$. The $|| \cdot ||$ represents the L2 norm or Euclidean norm of a vector.

*4.2.3 **Unbiased Visual Feature Representation**.* We aim to fuse two types of visual features of the sticker image (i.e., $v_p$ and $\overline{v}$) to obtain unbiased visual features. Specifically, we first simply concatenate $v_p$ with $\overline{v}$ as $[v_p; \overline{v}]$. Then, a linear layer is utilized to transform $[v_p; \overline{v}]$ into $d^r$-dimensional representation, denoted as unbiased visual features $v$. Finally, we facilitate the alignment of the emotional utterances representation with unbiased visual features for further consistent and accurate emotion recognition. Specifically, we consider the matched $(\overline{h}_T, v)$ pairs as positive samples, while other samples in the current batch $\mathcal{B}$ serve as negative samples. The contrastive loss [17] $\mathcal{L}_{co}$ is used for multimodal alignment.

### 4.3 Multimodal Encoder

We adopt a standard transformer encoder $\text{Enc}(\cdot)$ to integrate the multimodal features for information matching and emotion recognition. Following [49], we concatenate the dialogue context into a sequence and use BERT Embedding [8] layer to extract its features $\{x_i\}_{i=1}^{L_d}$, where $L_d$ is the number of tokens in the dialogue. To extract the textual information in the sticker, we incorporate PaddleOCR [10] to recognize text within the sticker and utilize BERT Embedding layer to obtain OCR feature $\{\hat{x}_i\}_{i=1}^{L_o}$, where $L_o$ is the length of the text. The multimodal encoder jointly processes the dialogue context, emotional utterances features, OCR features and unbiased visual features to obtain fusion features $X$:

$$X = \text{Enc}([E_{<\text{head}>}; x; \overline{h}_T; \hat{x}; v]), \tag{12}$$

where $X \in \mathbb{R}^{d^r}$ serve as the head token of output representation and $[;]$ denotes the concatenation operation. $E_{<\text{head}>}$ is the embedding of the '[HEAD]' token. We train an emotion classifier with the cross-entropy loss $\mathcal{L}_{em}$ to predict the speaker's emotion when selecting the sticker. Finally, we apply a fully-connected layer to produce the matching score of the positive or negative sticker. The multimodal information matching loss can be described as:

$$\mathcal{L}_{ma} = -\log P(y_m \mid X), \tag{13}$$

where the label $y_m$ indicates whether the dialogue information matches the sticker image, where a positive label is assigned the value of 1, while the negative label is 0. Note that for the negative sticker image, we directly employ the features $v_n$ extracted by CLIP as visual information input.

### 4.4 Total Loss

The objective of our CKS model is to minimize the total loss, i.e., emotion classification loss, contrastive loss and multimodal matching loss. The definition of total loss is:

$$\mathcal{L} = \frac{1}{D} \sum_{t=1}^{D} (\mathcal{L}_{ma} + \mathcal{L}_{co} + \lambda_1 \mathcal{L}_{em} + \lambda_2 \mathcal{L}_{tri}), \tag{14}$$

where $D$ is the total number of training samples, $\lambda_1$ and $\lambda_2$ stand for hyperparameters.

## 5 Experiment

### 5.1 Dataset

We evaluate our approach and conduct experiments on the English version of MOD [13] dataset. The dataset is derived from scenarios of daily conversational dialogues, which contain 307 stickers and 48 kinds of emotion labels. In accordance with [49], we divide each dialogue into multiple samples using the same preprocessing method. Each sample in the dataset comprises segments of the dialogue context along with an accompanying sticker. Specifically, there are 212,248 dialogues for training set, 3,183 for validation set, 3,189 for easy test set and 6,986 for hard test set. Notably, the hard test set includes stickers not encountered in the training set.

### 5.2 Experimental Details

We implement our method with Pytorch and train our model on 2 Tesla A100 40GB GPU cards. For the visual features extraction,

we use ViT-B/32 [9] to initialize the CLIP visual encoder. For bi-LSTM model, the dimension of the hidden state is set to $d$=300. The output feature dimension $d^r$ of CLIP visual encoder and knowledge-enhanced emotional utterance extractor is 768. The threshold of low and high frequency for the band-pass filter in Eq. 9 are 0.005 and 0.7 respectively. The margin parameter $\mu$ for the triplet loss is 0.2. Additionally, a transformer encoder model with 12 layers and a width of 512 is adopted for the multimodal encoder. During training, we adopt Adam optimizer [25] with an initial learning rate of 1e-5 and weight decay of 3e-4 to optimize the total loss function $\mathcal{L}$. We set the batch size to 64 and a maximum of 50 epochs. During inference, the number of candidate stickers $N$ is 11 in each dialogue. We combine contrastive scores with multimodal information matching scores for sticker selection. We set hyperparameters $\lambda_1$ and $\lambda_2$ for the loss function in Eq. 14 are 0.5 and 0.3, respectively.

## 5.3 Comparing Models

To verify the superiority of our CKS model, we first compare our method with six strong baseline models. The details of these baseline models are elaborated as follows: **CLIP** [37] and **ViLT** [24] are encoder-based vision-language models. We directly fine-tune them for sticker selection task. **MOD-GPT** [13] and **MMBERT** [49] is a language model based method to jointly process the sticker image and dialogue context. Furthermore, MMBERT is currently the state-of-the-art model in the sticker selection task. **BLIP-2** [29] and **FROMAGe** [26] are encoder-decoder based vision-language model. Considering FROMAGe utilizes a large language model (LLM) (i.e., OPT-6.7B [48]) and has pre-trained on a large scale of multimodal dialogue data, we directly apply it to the sticker selection task without additional fine-tunings.

## 5.4 Evaluation Metric

*5.4.1* **Automatic Evaluation Metrics**. Following previous research [15, 49], we evaluate the performance by considering the top 10 candidate stickers. We employ Recall at position $k$ in 10 candidates, denoted as $R_{10}@k$ ($k$=1, 3, 5), where $R_{10}@5$ is particularly suitable for practical application scenarios. We also utilize mean reciprocal rank ($MRR_{10}$) to measure the average of the reciprocal ranks of top-10 stickers. Besides, we also compute the emotion classification performance accuracy scores (Acc.).

*5.4.2* **Human Evaluation Criteria**. The automatic evaluation metrics are insufficient to evaluate the accuracy of the sticker selection since people frequently exhibit diverse sticker preference in online dialogues rather than a single correct sticker. To further refine our assessment, we invite 5 volunteers with good English education to perform a manual evaluation. We randomly select 200 samples from each model with the same dialogue id and ask volunteers to evaluate the quality of the ranking stickers using three criteria: Precision at position 3 and 5 for sticker selection ($\mathbf{P_s@3}$ and $\mathbf{P_s@5}$) evaluates whether the appropriate sticker exists in the top-3 and top-5 predicted stickers. Relevance to the top-3 emotional utterances at position 3 (**Rel-c**) assesses the degree to the top-3 stickers align with the leading three emotional utterances identified in the dialogue. Precision at position 3 for emotion classification ($\mathbf{P_e@3}$) evaluates whether the appropriate emotion exists in the top-5 prediction. Specifically, the **Rel-d** and **Rel-c** are scored on a scale from 0 to 2 (higher values indicate greater relevance), while $\mathbf{P_s@3}$, $\mathbf{P_s@5}$ and $\mathbf{P_m@3}$ are binary values for each sample. Table 3 shows the result of human evaluation. We choose CLIP, MOD-GPT and MMBERT as the comparison.

## 5.5 Results and Analysis

*5.5.1* **Performance Comparison**. Table 1 shows the automatic evaluation results on the MOD dataset. We observe that: **i)** For the validation and easy test set, our CKS outperforms all baseline models with significant margins on all metrics consistently. For example, our model improves the SOTA model MMBERT by "+10.24" on $R_{10}@1$, "+9.11" on $R_{10}@5$, "+11.82" on $MRR_{10}$ and "+5.75" on the accuracy of emotion classification in easy test set, showcasing its ability to select appropriate stickers in a way that resembles human emotional states. **ii)** By comparing the results in the hard test set, our model also achieves better performance over the baseline models on all metrics. In particular, CKS outperforms the MMBERT by "+7.29" and "+12.08" on the $MRR_{10}$ and accuracy of the emotion classification scores, respectively. This indicates that our model can more accurately comprehend the semantic information of utterances and the content of stickers, resulting in stronger generalization capabilities for selecting unseen stickers. **iii)** Notably, except for CLIP, all compared models incorporate dialogue history as input. Nonetheless, due to the absence of mining emotional utterances information within the dialogue history and its alignment with the unbiased visual features, they achieve lower scores than our model on all metrics. This clearly underscores the importance of emotional utterances within the dialogue context and unbiased visual features for sticker selection.

*5.5.2* **Ablation Study**. We conduct ablation experiments to verify the effectiveness of different components in our CKS model. Experimental results are shown in Table 2. In particular, **CKS w/o ITC**, **CKS w/o ITM**, **CKS w/o Emo** and **CKS w/o OCR** represent CKS without contrastive loss $\mathcal{L}_{co}$, multimodal information matching loss $\mathcal{L}_{ma}$, emotion classification loss $\mathcal{L}_{em}$ and OCR information for the multimodal encoder respectively. We find that: **i)** Compared to the results of CKS w/o KEUE and CKS, we observe that the emotional utterances information contributes to a significant improvement in sticker selection. e.g., Acc. increases from 48.76 to 52.73, $MRR_{10}$ increases from 48.97 to 54.86. It shows that the knowledge-enhanced emotional utterance extractor (KEUE) can effectively discern the emotional information in dialogues and select stickers that more closely align with human emotion. **ii)** We also investigate the effect of the visual knowledge deconfounder (VKD) and content invariant module (CIM) respectively. The results show that both modules in the unbiased visual feature extractor contribute an important improvement, which can alleviate the spurious correlations between sticker images and emotion, facilitating the sticker selection process. **iii)** By employing contrastive, multimodal information matching and emotion classification loss, our model also achieves an improvement on all metrics, e.g., "+4.98" on $MRR_{10}$ scores compared CKS with CKS w/o Emo. It shows the CKS model's capacity to facilitate a more refined alignment between emotional utterances representation and unbiased visual features.

**Table 1: Main automatic metrics results on validation set, easy test set and hard test set. Bold: the maximum value in the column. "-" indicates that these baseline models are primarily concentrated on the recall aspect of sticker selection, ignoring the emotion recognition during the sticker selection.**

| Model | Val | | | | | Easy Test | | | | | Hard Test | | | | |
|---|---|---|---|---|---|---|---|---|---|---|---|---|---|---|---|
| | $R_{10}@1$ | $R_{10}@3$ | $R_{10}@5$ | $MRR_{10}$ | Acc. | $R_{10}@1$ | $R_{10}@3$ | $R_{10}@5$ | $MRR_{10}$ | Acc. | $R_{10}@1$ | $R_{10}@3$ | $R_{10}@5$ | $MRR_{10}$ | Acc. |
| CLIP [37] | 21.90 | 47.72 | 66.04 | 41.00 | - | 22.26 | 48.41 | 66.07 | 40.52 | - | 16.96 | 39.54 | 58.88 | 35.73 | - |
| ViLT [24] | 17.31 | 40.00 | 63.27 | 35.37 | - | 20.86 | 49.95 | 70.09 | 41.65 | - | 14.55 | 32.82 | 56.32 | 32.16 | - |
| MOD-GPT [13] | 21.93 | 41.33 | 70.78 | 42.19 | 42.73 | 24.19 | 42.86 | 71.91 | 44.03 | 43.60 | 10.95 | 31.09 | 57.76 | 34.39 | 21.70 |
| MMBERT [49] | 23.97 | 40.75 | 73.45 | 44.58 | 45.27 | 25.65 | 43.74 | 76.17 | 46.61 | 51.09 | 11.43 | 34.41 | 59.63 | 36.58 | 29.69 |
| BLIP-2 [29] | 17.64 | 35.29 | 47.18 | 33.45 | - | 17.03 | 37.38 | 54.19 | 34.50 | - | 16.27 | 34.91 | 54.13 | 33.48 | - |
| FROMAGe [26] | 9.83 | 28.96 | 49.85 | 29.17 | - | 11.32 | 29.01 | 51.90 | 29.12 | - | 9.21 | 27.65 | 45.58 | 25.06 | - |
| CKS | **34.14** | **64.53** | **81.38** | **54.86** | **52.73** | **35.89** | **68.35** | **85.28** | **58.43** | **56.84** | **23.75** | **52.72** | **72.01** | **43.87** | **41.77** |

**Table 2: Ablation study results. Bold: the maximum value.**

| Method | $R_{10}@1$ | $R_{10}@3$ | $R_{10}@5$ | $MRR_{10}$ | Acc. |
|---|---|---|---|---|---|
| CKS w/o KEUE | 29.25 | 59.82 | 75.81 | 48.97 | 48.76 |
| CKS w/o VKD | 31.13 | 59.25 | 76.53 | 46.33 | 49.08 |
| CKS w/o CIM | 32.95 | 62.01 | 77.91 | 50.52 | 50.39 |
| CKS w/o ITC | 30.10 | 60.26 | 78.23 | 49.97 | 49.79 |
| CKS w/o ITM | 29.91 | 60.16 | 76.97 | 49.53 | 49.08 |
| CKS w/o Emo | 30.57 | 59.39 | 77.22 | 49.88 | - |
| CKS w/o OCR | 32.56 | 62.20 | 78.98 | 53.05 | 50.38 |
| CKS | **34.14** | **64.53** | **81.38** | **54.86** | **52.73** |

**Table 3: Human evaluation results. Bold: the maximum value.**

| Method | $P_s@3$ | $P_s@5$ | Rel-c | $P_m@3$ |
|---|---|---|---|---|
| CLIP [37] | 0.49 | 0.71 | 1.18 | - |
| MOD-GPT [13] | 0.47 | 0.68 | 0.92 | 0.58 |
| MMBERT [49] | 0.61 | 0.83 | 1.39 | 0.62 |
| CKS | **0.77** | **0.91** | **1.69** | **0.74** |

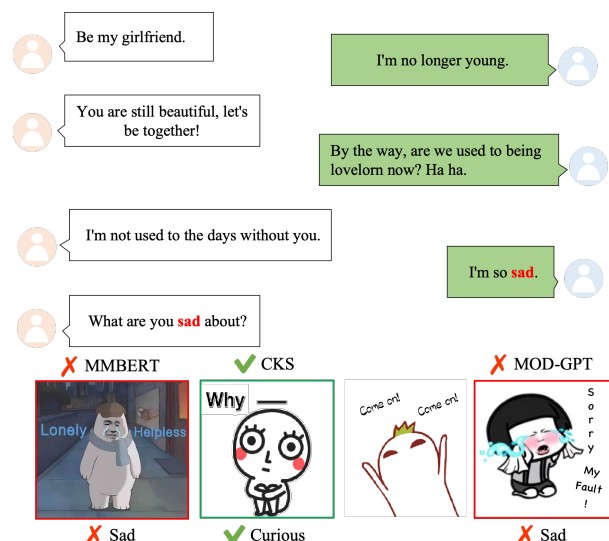

**Figure 6: A case study of selected stickers and predicted emotion by MOD-GPT, MMBERT and CKS.**

## 5.6 Case Study

Fig. 6 show selected stickers and predicted emotion by MOD-GPT, MMBERT and CKS. the CKS model is the only model that correctly selects the curious-themed sticker and predicts the correct emotion in Fig. 6. Specifically, although the co-occurrence of "sad-themed sticker + sad emotion label + word with sad emotion" / "word with sad emotion" = 41.33%, our CKS model selects the curious-themed sticker for the left speaker and recognizes the curious emotion, which further validates the effectiveness of our method to alleviates spurious correlations from dataset bias.

## 6 Conclusion

In this paper, we point out the spurious correlations in sticker selection are caused by the harmful dataset bias, i.e., frequent co-occurrence among the sticker images, words with emotional preference and emotion of the speaker. We propose the Causal Knowledge-Enhanced Sticker Selection (CKS) model to effectively mitigate such spurious correlations from a causal view. Specifically, we design a knowledge-enhanced emotional utterance extract to obtain emotional utterances and the interventional visual feature extractor for the alignment of emotional utterances representation with unbiased visual features. Extensive experiments on MOD dataset demonstrate the effectiveness of our CKS, which achieves state-of-art performance in the sticker selection task.

## Acknowledgments

This research is supported by the National Natural Science Foundation of China (62076100), the Fundamental Research Funds for the Central Universities, South China University of Technology (x2rjD2240100), the Science and Technology Planning Project of Guangdong Province (2020B0101100002), Guangdong Provincial Fund for Basic and Applied Basic Research—Regional Joint Fund Project (Key Project) (2023B1515120078), Guangdong Provincial Natural Science Foundation for Outstanding Youth Team Project (2024B1515040010), the China Computer Federation (CCF)-Zhipu AI Large Model Fund, the Hong Kong Polytechnic University's Postdoc Matching Fund (project no. P0049003).

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
