# OpenReview forum: "Deconfounded Emotion Guidance Sticker Selection with Causal Inference"
_acmmm.org/ACMMM/2024/Conference — MM2024 Poster_

### Official Review · Reviewer_hqEy · 2024-05-13

**Rating:** 4
**Confidence:** 4

**Summary:**

The manuscript is very well written, and the manuscript proposes a Causal Knowledge-Enhanced Sticker Selection model to mitigate spurious correlations. Specifically, the manuscript designs a knowledge-enhanced emotional utterance extractor to identify emotional information within dialogues. Then an interventional visual feature extractor is employed to obtain unbiased visual features, aligning them with the emotional utterance representation. Finally, a standard transformer encoder fuses the multimodal information for emotion recognition and sticker selection.

**Strengths:**

1. The abstractive cause descriptions are easier to understand and subsequently process.
2. The model is simple and intuitive.

**Limitations:**

But there are some problems, which must be solved before it is considered for publication. If the following problems are well-addressed, this reviewer believes that the essential contribution of this paper are important for multimodal sticker selection. Here are a few suggestions to enhance the overall standard of the work.

1. In Section "1 Introduction". The proposed "In light of pedagogy, the above limitations in the sticker selection model can be partially attributed to the model’s over-dependence on intuitive processing, rendering the models prone to various decision biases. The behavior of inappropriate sticker selection is referred to as the “heuristic system”. To address the limitation, human typically relies on an “analytic system” to select the appropriate sticker associated with emotion. "
Please present specific credible experimental results and analyses in the manuscript to show that the proposed method addresses this limitation.

2. In Section 3.1, sentence 6, "Ms" would be "Mc". Please maintain consistent between acronyms in the manuscript.

3. In Section "5.6 Case Study". Although this paper contains many case studies, the error cases of the CKS model are not presented to visualize the shortcomings of the CKS model.

4. In Section "6 Conclusion", please add the future directions in the field and the room for improvement of the proposed method.

5. Verify that all citations in the text are properly referenced, and vice versa. Ensure that the citation format is uniform and consistent.

**Suitability:**

3

---

### Official Review · Reviewer_N8aw · 2024-05-20

**Rating:** 4
**Confidence:** 3

**Summary:**

The authors propose a Causal Knowledge-Enhanced Sticker Selection model to mitigate spurious correlations, referred to as CKS. The model has three modules: the first part is the knowledge enhanced emotional utterance extractor to identify emotional utterances with common sense knowledge. The second part is an intrusive visual feature extractor to obtain unbiased visual features and align them with the emotional discourse representation obtained in the first part. And the third part uses a multimodal encoder to process multimodal information. Finally, experiments on MOD data sets verify the effectiveness of the model.
There are some strong points and weak points as follows.

**Strengths:**

1.	The research problem is important. The authors present a causal graph to analyze the cause of the spurious correlations in sticker selection and use the Causal Knowledge-Enhanced Sticker Selection model to mitigate spurious correlations.
2.	The role of each module is explained clearly by combining the causal graph and the model framework graph, and various spurious correlations are mitigated to a certain extent.
3.	The evaluation mechanisms are diverse and persuasive in terms of model effectiveness.

**Limitations:**

1.	The introduction section lacks specific data support to demonstrate that general models will learn this kind of spurious correlation, rather than just a few special examples.
2.	The authors seem to emphasize the impact of words with specific emotional biases on the model's judgments and mitigate this spurious correlation through the visual knowledge deconfounder (VKD) module. However, the employed method lacks novelty.
3.	The authors provide examples to illustrate the significant role of the model in mitigating spurious correlations between variables V and Y. However, no examples are given regarding how the emotional utterance extractor and content-invariant module would affect the model's judgments.

**Suitability:**

3

---

### Official Review · Reviewer_xUX7 · 2024-05-24

**Rating:** 2
**Confidence:** 3

**Summary:**

For the sticker selection task, this paper introduces a causal graph to explicitly identify spurious correlations in sticker selection tasks, proposing a Causal Knowledge-Enhanced Sticker Selection (CKS) model to mitigate such biases. Combining a knowledge-enhanced emotional utterance extractor and an interventional visual feature extractor, it achieves unbiased emotion recognition and sticker selection. The effectiveness of the CKS is evaluated on the MOD dataset.

**Strengths:**

-	This paper deals with the interesting sticker selection task and first investigates the inherent bias in the sticker selection task from a causal inference perspective.

-	The ablation of the usefulness of every used component is quite detailed.

**Limitations:**

-	The experiment was only conducted on one dataset. Gao et al. proposed a large-scale dialogue-sticker selection dataset StrickerChat [1], offering a richer collection of stickers than MOD. It is recommended to supplement the results on StrickerChat to validate the generalizability of CKS.
-	It is noted that the performance of some comparison methods differs greatly from those reported in the original paper, such as MOD-GPT and MMBERT. It is essential to explain why such differences occur. Furthermore, it is advisable to include the missing metrics in Table 1 for a more comprehensive comparison.
-	In addition to the reported metrics, it would be beneficial to validate the performance improvement resulting from mitigating spurious correlations more intuitively. Relying solely on a single case for demonstration lacks persuasiveness. Moreover, in Section 5.6, it is mentioned that "'sad-themed sticker + sad emotion label + word with sad emotion' / 'word with sad emotion' = 41.33%". However, the manuscript does not seem to explain how this statistic is obtained, and it would be helpful to show the complete statistic results for better understanding.

[1] Gao S, Chen X, Liu C, et al. Learning to respond with stickers: A framework of unifying multi-modality in multi-turn dialog. WWW 2020.

**Suitability:**

3

---

### Official Review · Reviewer_GKDe · 2024-05-26

**Rating:** 4
**Confidence:** 3

**Summary:**

This paper investigates the sticker selection task. In this work, the authors propose a Causal Knowledge-Enhanced Sticker Selection model to mitigate spurious correlations. Specifically, they design a knowledge-enhanced emotional utterance extractor to identify emotional information within dialogues. Then, an interventional visual feature extractor is employed to obtain unbiased visual features, aligning them with the emotional utterances representation. Finally, a standard transformer encoder fuses the multimodal information for emotion recognition and sticker selection.

**Strengths:**

The task and the proposed method are interesting.

The paper is well-organized and easy to follow.

**Limitations:**

- The caption of Figure 3 needs to be further improved. For example, the abbreviation of VKD needs clarification.

- In the Ablation Study, I care about the experiments that not using knowledge. Knowledge is considered as the additional information, and it is unfair to compare it with other methods.

**Suitability:**

2

---

### Meta-Review · Area_Chair_P83p · 2024-06-30

**Recommendation:** Accept (Poster)
**Confidence:** 4

**Metareview:**

This paper studies the sticker selection task and proposes a Causal Knowledge-Enhanced Sticker Selection model to mitigate spurious correlations. The authors design a knowledge-enhanced emotional utterance extractor to identify emotional information within dialogues. An interventional visual feature extractor is then employed to obtain unbiased visual features, aligning them with the emotional utterance representations. A standard transformer encoder then fuses the multimodal information for emotion recognition and sticker selection.  Experiments on MOD datasets show the effectiveness of the method.

All reviewers agree that the task and the proposed method are interesting, and the paper is well-organized and easy to follow. Several concerns regarding the experiment, e.g., the baselines and the metrics, are addressed in the rebuttal. I strongly encourage the authors to incorporate the discussions and carefully address the comments from the reviews in the final version.